**Brief Communication**

# Improving atlas-scale single-cell annotation models with hierarchical cross-entropy loss

Sebastiano Cultrera di Montesano [1,8] ✉, Davide D'Ascenzo [2,3,8], Srivatsan Raghavan [1,4,5,6,9], Ava P. Amini [7,9], Peter S. Winter [1,9] ✉ & Lorin Crawford [7,9] ✉

Accurately annotating cell types is essential for extracting biological insight from single-cell RNA sequencing data. Although cell types are naturally organized into hierarchical ontologies, most computational models do not explicitly incorporate this structure into their training objectives. Here, we introduce a hierarchical cross-entropy loss that aligns model objectives with biological structure. Applied to architectures ranging from linear models to transformers, this simple modification improves out-of-distribution performance by 12–15% without added computational cost. Critically, we underscore the need to focus on new data generation that improves the connectivity among annotated cell types. Our work suggests that this is likely to yield more generalizable algorithms than would solely increasing model complexity.

Cell-type annotation is a core step in single-cell RNA sequencing (RNA-seq) pipelines. The quality of annotations directly impacts downstream analyses, including mapping cellular diversity across tissues and deciphering cell-type-specific regulatory mechanisms. Manual annotation remains time-consuming and dependent on domain-specific expertise, but the rapid adoption of single-cell RNA-seq as a standard laboratory technique has created an urgent need for automated, scalable solutions[1]. With repositories such as the Human Cell Atlas[2] and CELLxGENE[3] now containing over 100 million cells, accurate and robust annotation methods are a critical first step in translating these large-scale datasets into actionable biological insights[4,5].

Automated atlas-level cell-type annotation can be framed as a supervised classification problem, where models assign labels to individual cells on the basis of gene expression profiles, using reference annotations provided by original studies[6–8]. A defining feature of this task is that cell types are organized within a hierarchical ontology[9,10], forming a multilevel taxonomy. For example, 'leukocytes' represent a broad category that contains 'lymphocytes', which in turn includes more specific subtypes such as 'B cells'. However, annotation practices vary substantially between studies—some assign broad categories,

while others distinguish fine-grained subtypes. This inconsistency in label granularity introduces ambiguity into the training signal, as models must infer the appropriate level of resolution without explicit guidance. More formally, the annotation task can be viewed as learning a function $f: X \rightarrow Y$, where $X$ is the space of gene expression profiles and $Y$ is a structured label space defined by a directed acyclic graph (DAG). In this graph, each node corresponds to a cell type and directed edges represent subtype relationships—for example, 'B cell' and 'T cell' are children of 'lymphocyte'. This structure captures relationships across varying levels of annotation granularity[11–13].

Many methods have been developed to perform automated cell-type annotation, ranging from logistic regression to deep learning architectures[14–17]. Recent benchmarking studies have shown that deep learning models outperform simpler methods as the number of cells in a dataset increases[11]. Importantly, these evaluations were conducted using donor-partitioned training and test splits, a design we refer to as the in-distribution (ID) setting (Fig. 1a). While useful for controlled comparisons, such splits do not reflect how cell atlases evolve in practice, where new studies are continually added and must be annotated upon release.

[1]Broad Institute of MIT and Harvard, Cambridge, MA, USA. [2]Politecnico di Torino, Turin, Italy. [3]Università degli Studi di Milano, Milan, Italy. [4]Department of Medical Oncology, Dana-Farber Cancer Institute, Boston, MA, USA. [5]Harvard Medical School, Boston, MA, USA. [6]Department of Medicine, Brigham and Women's Hospital, Boston, MA, USA. [7]Microsoft Research, Cambridge, MA, USA. [8]These authors contributed equally: Sebastiano Cultrera di Montesano, Davide D'Ascenzo. [9]These authors jointly supervised this work: Srivatsan Raghavan, Ava P. Amini, Peter S. Winter, Lorin Crawford. ✉e-mail: scultrer@broadinstitute.org; pwinter@broadinstitute.org; lcrawford@microsoft.com

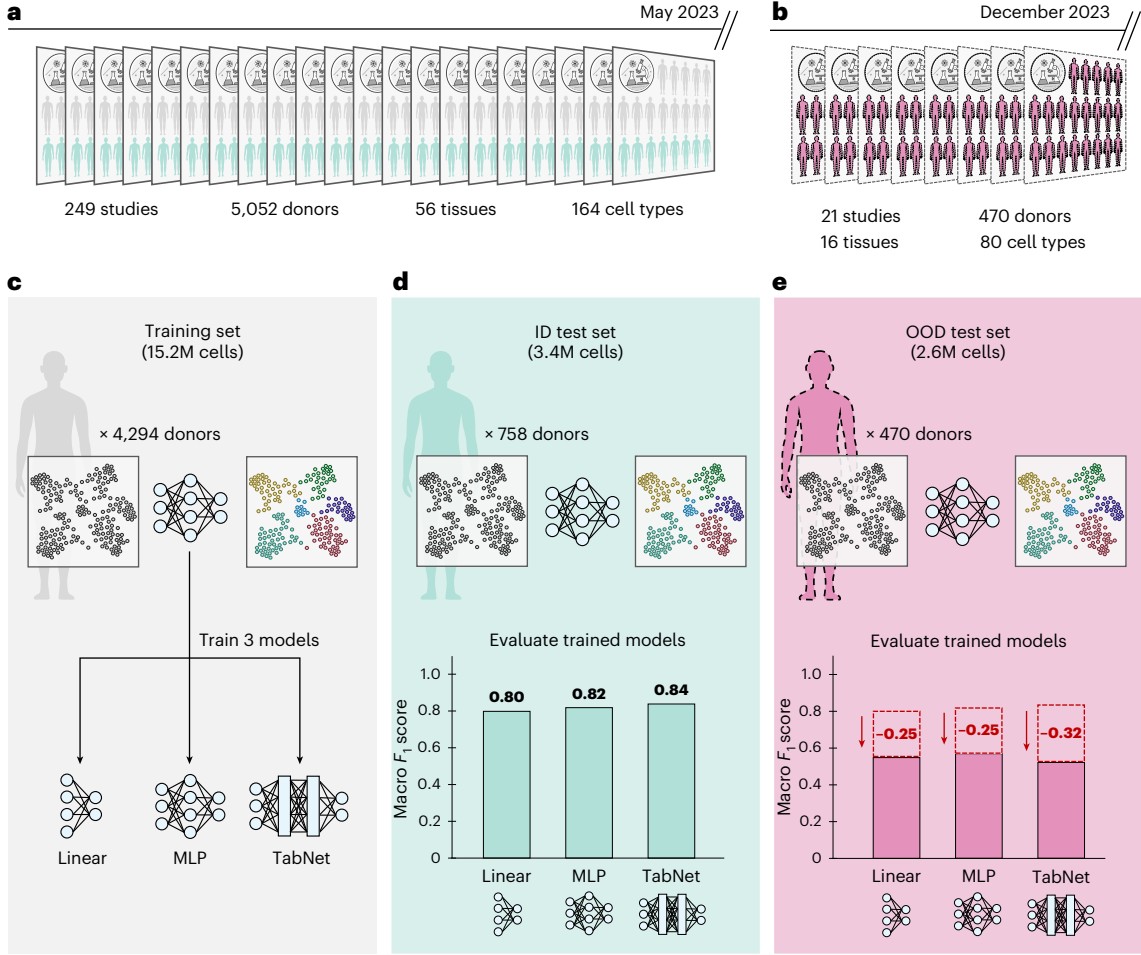

**Fig. 1 | Evaluating model generalization in continuously updated single-cell atlases reveals sharp OOD performance drops for the annotation task.**
**a**, A curated subset of the CELLxGENE census (May 2023 release) consisting of 22.2 million human cells annotated with 164 curated cell types, spanning 5,052 donors, 56 tissues and 249 studies. All cells were profiled using 10x Genomics platforms. **b**, OOD test set comprised of 2.6 million newly added cells from 21 studies in the December 2023 release. These cells span 470 donors and 16 tissues, and are annotated with 80 of the 164 original training cell types. All cells are also profiled using 10x Genomics platforms. **c**, We train three models (linear classifier,

MLP and TabNet) on a donor-partitioned training set comprised of 15.2 million cells from the May 2023 CELLxGENE census. **d**, ID test set comprised of 3.4 million cells from the May 2023 release of the CELLxGENE census, held out by donor. The linear model, MLP and TabNet achieve 80%, 82% and 84% macro $F_1$ scores, respectively (mean taken over four independent runs). **e**, All models exhibit substantial OOD performance drops: macro $F_1$ scores decrease to 55%, 57% and 52% for the linear model, MLP and TabNet, respectively. The dashed red bars indicate the ID performances for comparison.

## Results

To better evaluate generalization to newly released studies, we consider an out-of-distribution (OOD) set-up in which models are tested on datasets not seen during training (Fig. 1b). We trained three methods with increasingly complex architectures (a linear classifier, a multilayer perceptron (MLP) and TabNet[18]) on an atlas of 15.2 million human cells annotated with 164 unique cell types, curated in the scTab study[11] from the May 2023 release of the CELLxGENE census (Fig. 1c). We then evaluate each method on 2.6 million human cells from 21 studies newly added during the 2023-12-15 release, spanning 470 donors, 16 tissues and 80 of the original 164 cell types represented in the training set. Despite being evaluated on the same cell types profiled with the same assays, macro-averaged $F_1$ (macro $F_1$) scores dropped by 24–32% for the linear classifier, MLP and TabNet when moving from the ID case (Fig. 1d) to the OOD setting (Fig. 1e), underscoring the limitations of current modeling strategies in generalizing across studies.

To address these shortcomings, we introduce a hierarchical cross-entropy (HCE) loss that explicitly incorporates the structural relationships between cell types (Methods). Unlike standard cross-entropy (CE), which treats all classes as flat and independent, HCE enforces a

consistency constraint: the probability assigned to a general cell type (for example, T cell) must be at least as high as the sum of more granular subtypes (for example, 'α–β T cell' and 'γ–δ T cell'). This prevents the model from needing to choose between broad and granular labels, since predicting a child inherently implies selecting the parent in the hierarchy (Fig. 2a).

While there are methods that leverage ontological information for cell-type annotation, they do not enforce hierarchical consistency as an integral part of their predictive framework. For example, OnClass maps both transcriptomic profiles and cell ontology structure into a joint embedding space, enabling both the annotation of unseen cell types and the identification of marker genes[12]. However, it operates primarily as a nearest-neighbor or embedding search algorithm and does not couple hierarchical relationships to the learned probabilities for each cell. As a result, sibling classes or intermediate states can still be misassigned if their embeddings overlap in feature space. As another example, popV aggregates predictions from multiple classifiers using ontology-based voting, producing robust consensus labels and uncertainty estimates for ambiguous or outlier populations[7]. However, the ontology is used only as a scaffold for post hoc reconciliation and not as a guide for

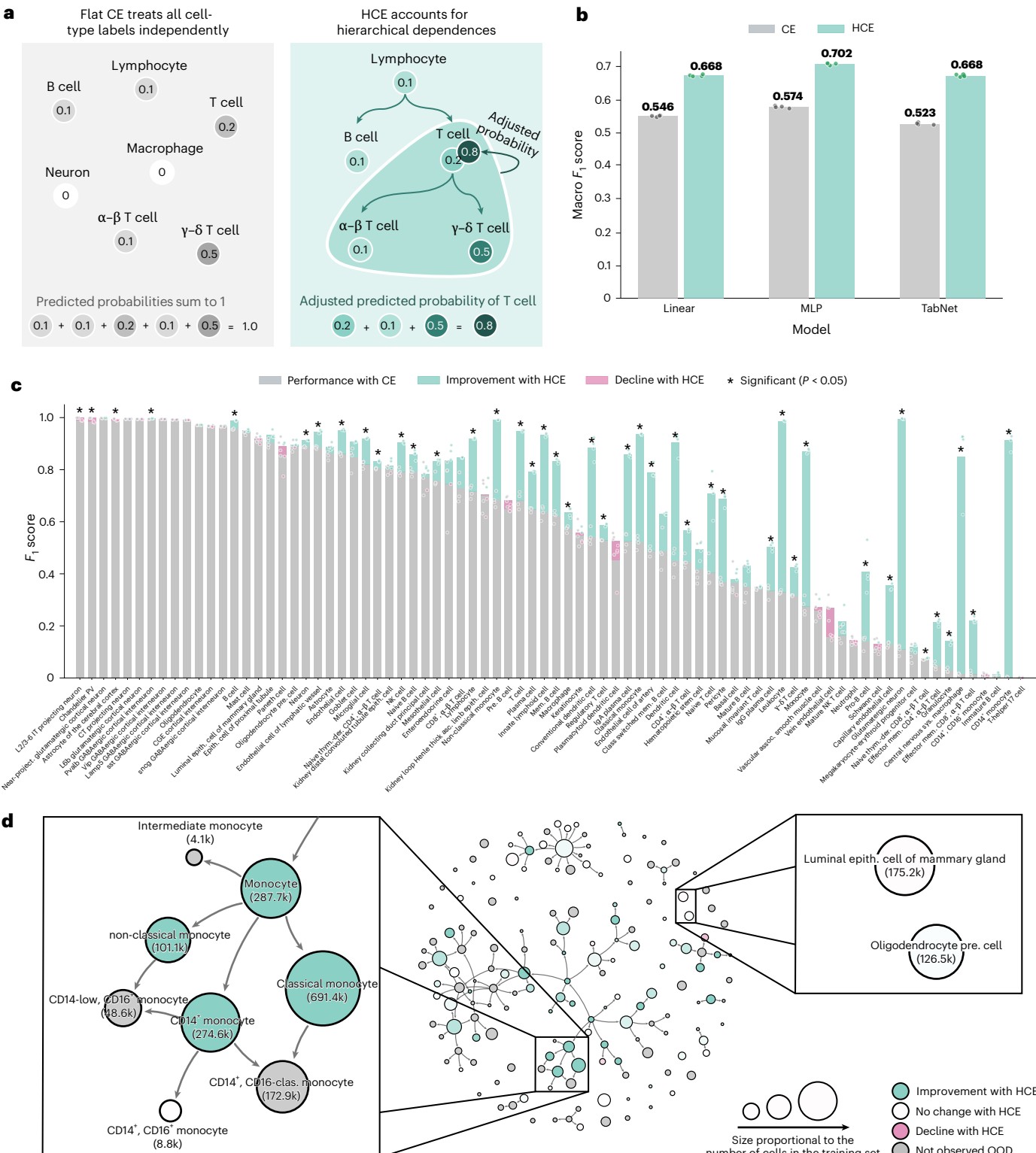

**Fig. 2 | HCE loss improves perfomances across architectures. a**, The standard CE loss defines a probability distribution over a flat label set, treating each cell type independently and requiring that probabilities sum to unity across the ontology. The HCE loss modifies these predictions by propagating probability mass up the DAG of the cell ontology: parent nodes such as T cell accumulate mass from their more specific descendants, such as α−β T cell and γ−δ T cell, encouraging biologically coherent predictions. **b**, The HCE loss improves macro $F_1$ scores by 12–15% on OOD evaluations across the linear classifier, MLP and TabNet. All performance metrics reported reflect the mean taken over four independent training and evaluation runs per model, with results from each run shown as individual dots (color coding remains the same as in the legend). **c**, Per-cell-type performance changes induced by the HCE loss strategy for the MLP model, shown relative to standard CE. All performance metrics reported reflect the mean taken over four independent training and evaluation runs per model, with results from each run shown as individual dots (color coding remains the same as in the legend). For each cell type, a paired $t$-test was performed and $P$ values were adjusted using the Holm–Bonferroni method to correct for multiple hypothesis testing. **d**, Improvements from HCE loss for the MLP model visualized directly on the cell ontology DAG consisting of all 164 cell types seen in the training set. Node size reflects the number of cells of that type seen in training, while color indicates the change in $F_1$ score, shown as a gradient from green (improvement) to white (neutral/no change) to red (decline). Gray nodes correspond to cell types not observed in the OOD test set.

model optimization. This means that hierarchical constraints are not encoded in training and possible conflicts or inconsistencies in the ensemble are resolved heuristically. In contrast, SCimilarity focuses on metric learning for scalable, cross-study retrieval of transcriptionally similar cells, using the ontology at training time to exclude ambiguous annotation pairs when sampling triplets for a contrastive loss function[19]. The learned representation supports high-quality search and transfer tasks but is not directly optimized for hierarchical or taxonomic consistency when determining class probabilities. In summary, unlike these approaches, our HCE loss explicitly encodes hierarchical dependences and relationships into the model's objective, ensuring that all predictions respect the structure of the cell ontology.

Applying the HCE loss improved OOD macro $F_1$ scores by 12–15% for the linear classifier, MLP and TabNet, without modifying their architecture or tuning any hyperparameters (Fig. 2b). These consistent gains demonstrate the widespread benefits of hierarchy-aware training for cell-type annotation tasks. The HCE loss function enables the recovery of roughly half of the performance drop observed when models are applied to new studies, underscoring the practical value of aligning training objectives with ontology structure. To further assess the consistency of this effect, we evaluated performance across each of the 21 held-out studies individually (Supplementary Fig. 1). Outside of just one study while using the linear model, all HCE-trained models showed statistically significant improvements, highlighting the robustness of this approach across diverse experimental settings.

To better understand where the improvements from hierarchical training arise, we identified cell types that exhibited statistically significant changes in performance between models trained with and without the HCE loss. These differences were determined using a paired $t$-test across training runs, with $P$ values corrected using the Holm–Bonferroni method (Methods). In the MLP model, for example, HCE led to improvements of up to 0.9 in $F_1$ score for cell types such as 'glutamatergic neuron' and 'CD14$^+$ monocyte' (Fig. 2c). Examining these effects in relation to the cell ontology, we found that the largest gains occurred for internal nodes, particularly those embedded in densely connected regions of the DAG where related types were annotated in the training data (Fig. 2d and Supplementary Figs. 2–4). In contrast, leaf nodes—especially structurally isolated ones—showed more modest gains (Supplementary Figs. 5 and 6). This aligns with the intuition that the hierarchical loss is most effective when it can propagate signal across nearby cell types. Similar trends were observed for the linear and transformer-based models, highlighting the architecture-agnostic nature of the effect (Supplementary Fig. 7). Importantly, gains were largely unaffected by a cell type's rarity, the number of contributing studies or tissues and the diversity of sequencing technologies used—further underscoring the robustness of the approach (Supplementary Fig. 8). Finally, these gains extend to cells observed in new contexts, including across diseases and tissues not seen in the training set, where we also observe consistent improvements (Supplementary Figs. 9 and 10).

## Discussion

Our results challenge the view that increasing model complexity is the primary route to improved cell-type annotation at atlas scale. Instead, we demonstrate that aligning the training objective with biological structure—through an HCE loss—consistently improves generalization across model classes, from linear classifiers to transformers. Critically, our findings suggest a strategy for building more effective training sets: rather than simply adding data, efforts should prioritize studies that increase connectivity among annotated cell types, especially in sparsely represented regions, thereby amplifying the generalization capabilities of learning architectures. While the hierarchical objective recovers roughly half of the OOD performance drop, some decrease in accuracy is expected due to imperfect annotation agreement across studies. Understanding the downstream

impacts of these inconsistencies and correcting them will be essential as atlas-level resources become more standardized and begin to power the next generation of AI-driven discoveries in biology. It is important to note that HCE relies on a predefined, labeled DAG, and while the cell ontology serves as a valuable reference it is continuously evolving, with ongoing updates to cell-type definitions and their hierarchical relationships. Furthermore, while this study centers on cell-type classification, the hierarchical loss generalizes to any setting with structured label spaces, offering a simple drop-in replacement for standard CE that brings domain knowledge into model training. This points to a broader opportunity to incorporate biological priors into learning objectives, an increasingly important consideration as models are trained on ever-growing single-cell atlases[19–21].

## Methods

### Training and evaluation datasets

The dataset used in this study originates from the same filtered subset of the CELLxGENE census (v.2023-05-15)[3] that was curated for the scTab study[11]. This subset was constructed by applying strict inclusion criteria to the full census: only primary human cells profiled with 10x Genomics technologies were retained and the feature space was limited to 19,331 human protein-coding genes. Cell types were required to appear in at least 5,000 cells drawn from a minimum of 30 donors. All gene expression profiles were size-factor normalized to 10,000 counts per cell and log-transformed with a pseudocount of 1 (that is, $f(x) = \log(x + 1)$). The resulting dataset included 22,189,056 cells annotated with 164 distinct cell types, spanning 5,052 donors and 56 tissues. For the ID task, we adopted the same donor-partitioned data split as used by Fischer et al.[11]—that is, 15,240,192 cells for training, 3,500,032 for validation and 3,448,832 for testing.

The OOD test dataset consisted of all newly added human cells in a subsequent release of the CELLxGENE census (v.2023-12-15). These cells were also profiled using 10x Genomics platforms and annotated with one of the 164 labels observed during training. This resulted in approximately 2.6 million cells drawn from 21 studies, covering 80 of the 164 training cell types.

### Cell ontology

We used the cell ontology obtained from the Ontology Lookup Service at EMBL-EBI as the hierarchical scaffold for all analyses[10]. The ontology was represented as a DAG, where nodes correspond to cell types and directed edges correspond to `is_a` subtype relationships. We restricted the ontology to the 164 distinct cell types observed in the training set ('Training and evaluation datasets'). In CELLxGENE, which is the atlas used in our study, cell types are annotated by the original data contributors and then harmonized by mapping each label to the closest cell ontology term as specified by the portal's data schema. While the cell ontology offers a valuable scaffold for representing hierarchical relationships among cell types, it is important to note that its structure is continuously being revised where certain definitions and mappings between cell types remain under active refinement.

Because each cell type corresponds to a node in the DAG, we can further classify them on the basis of the type of node they represent. A node was defined as a leaf if it had no children in the pruned ontology and as an internal node if it had at least one child. We also distinguished between connected nodes, which had at least one parent or child present in the curated training set, and isolated nodes, which had none of their ancestors or descendants represented in the training data. These definitions were used to assess how the hierarchical loss propagates information across the ontology (Supplementary Fig. 6).

### Evaluation protocol

Classification performance was evaluated using the macro $F_1$ score, which computes the unweighted average of the $F_1$ scores across all cell types. This metric ensures that each cell type contributes equally

to the overall score, regardless of class imbalance or prevalence in the dataset. For $C$ cell types, the macro $F_1$ score is computed as

$$\text{macro } F_1 \text{ score} = \frac{1}{C} \sum_{i=1}^{C} \frac{2 \times \text{precision}_i \times \text{recall}_i}{\text{precision}_i + \text{recall}_i} \qquad (1)$$

where precision$_i$ and recall$_i$ are defined for the $i$th class as

$$\text{precision}_i = \frac{\text{TP}_i}{\text{TP}_i + \text{FP}_i}, \qquad \text{recall}_i = \frac{\text{TP}_i}{\text{TP}_i + \text{FN}_i}. \qquad (2)$$

Here, the terms TP$_i$, FP$_i$ and FN$_i$ denote the number of true positives, false positives and false negatives for the $i$th cell type, respectively. We followed the evaluation framework introduced by Fischer et al.[11] in the scTab study, particularly because of the way those authors handled differences in the granularity of annotations that can occur across different studies: namely, a predicted label is considered correct if it exactly matches the ground-truth label or if it corresponds to a descendant of the ground-truth label in the cell ontology (that is, the prediction is a more specific subtype). This accounts for the fact that some datasets provide coarse-grained annotations (for example, T cell) while others include more detailed subtypes (for example, 'CD4-positive, α–β T cell'). In such cases, predicting a valid subtype is treated as correct, as it remains consistent with the original label. Any other prediction, including a coarser label (such as a parent node) or an unrelated class, is considered incorrect.

## Model details

We evaluated three model architectures of increasing complexity: a linear classifier, an MLP and the TabNet transformer model. Each model takes as input the full set of 19,331 human protein-coding genes. To ensure a fair comparison across models and with previous work, we adopted the architecture configurations and hyperparameters used in the scTab benchmarking study from Fischer et al.[11] (Supplementary Tables 1–3). The models using CE versus HCE share identical architecture and hyperparameter settings; the loss term is the only difference between them. Specifically, for the models with CE, we used the best hyperparameters available according to the original scTab study. For the models using the HCE loss, we did not perform additional hyperparameter tuning and instead kept the (possibly suboptimal) hyperparameters used for the models with CE. Note that, while recent efforts have explored large-scale foundation models to learn transferable embeddings for single-cell data, such approaches have not yet demonstrated clear advantages over simpler, task-specific approaches for cell-type annotation[11,22]. We therefore focused on methods where we could easily isolate and study the direct effects of implementing the HCE strategy.

## HCE loss function

The HCE loss function extends the standard CE loss by explicitly encoding the structural relationships across the cell ontology. With the standard CE, the loss is computed directly from raw model predictions, treating all cell types as independent classes. Let $\mathbf{p} = (p_1, \ldots, p_C)$ denote the raw predicted probabilities for $C$ different cell types. The standard CE loss is given by

$$\mathcal{L}_{\text{CE}} = -\sum_{i=1}^{C} \mathbb{1}\{\text{label} = i\} \log p_i \qquad (3)$$

where $\mathbb{1}\{label = i\}$ is an indicator function that is equal to 1 if the true class label is the $i$th cell type and 0 otherwise. The HCE adjusts these predictions to reflect hierarchical dependences encoded in the ontology's DAG. The adjusted score $s_i$ for the $i$th cell type is computed as the sum of the predicted probability for its label and the predicted probabilities of all its descendant subtypes

$$s_i = p_i + \sum_{j \in \mathcal{D}(i)} p_j \qquad (4)$$

where $\mathcal{D}(i)$ denotes the set of all descendants of cell type $i$ in the DAG. This adjustment ensures that the probability of a parent node reflects its entire subgraph. The hierarchical loss is then

$$\mathcal{L}_{\text{HCE}} = -\sum_{i=1}^{C} \mathbb{1}\{\text{label} = i\} \log s_i. \qquad (5)$$

This formulation directly parallels the evaluation framework, where predictions are considered correct if they match the ground-truth label or any of its descendants. By aligning the training objective with the assessment criterion, HCE encourages cell-type classification models to distribute probability mass in a way that respects biological hierarchy and annotation granularity.

Consider an ontology subgraph that is rooted at the node T cell, which includes subtype labels such as CD4$^+$ T cell, CD8$^+$ T cell and γ–δ T cell. The HCE enables classifications models to predict fine-grained subtypes when available, while also deferring to parent categories when annotations are coarse or ambiguous. For example, if some studies annotate cells as T cell while others use more specific labels such as CD4$^+$ T cell or CD8$^+$ T cell, the adjusted score is computed as

$$s_{\text{Tcell}} = p_{\text{Tcell}} + p_{\text{CD4+}} + p_{\text{CD8+}} + p_{\gamma-\delta} + \ldots . \qquad (6)$$

This hierarchical set-up allows the model to aggregate subtype information upward, improving consistency across annotations with varying granularity.

## Implementation details for the HCE loss

We implemented the HCE loss using a reachability matrix $R \in \{0, 1\}^{C \times C}$, where element $R_{ij} = 1$ if the $j$th class is reachable from the $i$th class (meaning $j$ is either $i$ itself or $j$ is a descendant of $i$ in the hierarchy) and $R_{ij} = 0$ otherwise. The reachability relation encoded in this matrix is a partial order and has the following mathematical properties:

- Reflexive—every class is reachable from itself (diagonal elements are 1).
- Antisymmetric—if class $i$ can reach $j$ and $j$ can reach $i$, then $i = j$.
- Transitive—if class $i$ can reach $j$ and $j$ can reach $k$, then $i$ can reach $k$.

Indeed, the reachability matrix represents the transitive closure of the inverted adjacency matrix of the hierarchical DAG structure. Since the original DAG encodes `is_a` relationships from child to parent, we invert the edge directions to enable parent-to-descendant reachability, ensuring reflexivity by setting the diagonal to 1. Each trained model outputs a raw probability distribution $\mathbf{p} = (p_1, \ldots, p_C)$ over the class labels. The adjusted scores are computed via matrix–vector multiplication: $\mathbf{s} = R\mathbf{p}$, which efficiently aggregates descendant probabilities for each class. We then apply a log transformation with numerical stability $\log(\mathbf{s} + \epsilon)$, where $\epsilon = 10^{-6}$. The final loss uses a weighted negative log-likelihood as implemented in PyTorch, with class weights computed following scikit-learn's `compute_class_weight` approach: $w_i = N/(Cn_i)$, where $N$ is the total number of samples, $C$ is the number of classes and $n_i$ is the count of samples for the class $i$. The complete loss for a single training sample $x$ with true label $t$ is

$$\mathcal{L}_{\text{HCE}}(x) = -w_t \log(s_t + \epsilon). \qquad (7)$$

This formulation maintains consistency with the models trained with the weighted CE, while incorporating hierarchical structure through efficient matrix operations.

## Statistical evaluation of performance differences across loss functions

To assess changes in predictive performance induced by the ontology-aware training strategy, we computed per-cell-type

differences in macro $F_1$ score between models trained with standard CE and HCE across four independent training runs. For each cell type, a paired *t*-test was performed and *P* values were adjusted using the Holm–Bonferroni method to correct for multiple hypothesis testing. Statistically significant differences indicate cell types for which ontology-aware training produces consistent changes beyond random variability.

### Reporting summary
Further information on research design is available in the Nature Portfolio Reporting Summary linked to this article.

## Data availability
The datasets used in this work were obtained from CELLxGENE (census v.2023-12-15). A preprocessed version of the data has been made available by the scTab study[11]: https://pklab.med.harvard.edu/felix/data/merlin_cxg_2023_05_15_sf-log1p.tar.gz. Ontology relationships were resolved using the Ontology Lookup Service: https://www.ebi.ac.uk/ols/ontologies/cl. Model checkpoints needed to reproduce the results in this work can be found via Zenodo at https://zenodo.org/records/17211022 (ref. 23). Source data for Figs. 1d,e and 2b,c,d and Supplementary Figs. 1–4, 7, 9 and 10 are available with this Brief Communication.

## Code availability
The source code for HCE loss is available under the MIT license via GitHub at https://github.com/microsoft/hce-classification (ref. 24).

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

## Acknowledgements
We thank E. Forte for her careful revision of the paper and for insightful suggestions on improving the clarity and visual presentation of the figures. S.C.d.M. was supported by the Eric and Wendy Schmidt Center at the Broad Institute of MIT and Harvard. D.D. was financially supported by the Italian National PhD Program in Artificial Intelligence (DM 351 intervento M4C1—Inv. 4.1—Ricerca PNRR), funded by NextGenerationEU (EU-NGEU). This research was supported in part by a David & Lucile Packard Fellowship for Science and Engineering awarded to L.C. S.R. acknowledges funding support from NCI K08 CA260442. Any opinions, findings and conclusions or recommendations expressed in this material are those of the author(s) and do not necessarily reflect the views of any of the funders.

## Author contributions
S.C.d.M., D.D. and L.C. jointly designed the study. S.C.d.M. and D.D. implemented the method, performed the analyses and wrote the initial draft of the paper. S.R., P.S.W., A.P.A. and L.C. supervised the project and provided resources. All authors contributed to interpreting the results and revising the paper.

## Competing interests
S.R. holds equity in Amgen. S.R. and P.S.W. receive research funding from Microsoft. A.P.A. and L.C. are employees of Microsoft and own equity in Microsoft. S.C.d.M. reports compensation from Engine Ventures and Mobius Biotechnology GmbH for consulting/speaking unrelated to this work. P.S.W. reports compensation from Engine Ventures and AbbVie for consulting/speaking unrelated to this work. The other authors have no competing interests.

## Additional information

**Correspondence and requests for materials** should be addressed to Sebastiano Cultrera di Montesano, Peter S. Winter or Lorin Crawford.

# Reporting Summary

## Statistics

For all statistical analyses, confirm that the following items are present in the figure legend, table legend, main text, or Methods section.

| n/a | Confirmed | |
|---|---|---|
| ☐ | ☒ | The exact sample size (*n*) for each experimental group/condition, given as a discrete number and unit of measurement |
| ☐ | ☒ | A statement on whether measurements were taken from distinct samples or whether the same sample was measured repeatedly |
| ☐ | ☒ | The statistical test(s) used AND whether they are one- or two-sided<br>*Only common tests should be described solely by name; describe more complex techniques in the Methods section.* |
| ☐ | ☒ | A description of all covariates tested |
| ☐ | ☒ | A description of any assumptions or corrections, such as tests of normality and adjustment for multiple comparisons |
| ☐ | ☒ | A full description of the statistical parameters including central tendency (e.g. means) or other basic estimates (e.g. regression coefficient) AND variation (e.g. standard deviation) or associated estimates of uncertainty (e.g. confidence intervals) |
| ☒ | ☐ | For null hypothesis testing, the test statistic (e.g. *F*, *t*, *r*) with confidence intervals, effect sizes, degrees of freedom and *P* value noted<br>*Give P values as exact values whenever suitable.* |
| ☒ | ☐ | For Bayesian analysis, information on the choice of priors and Markov chain Monte Carlo settings |
| ☐ | ☒ | For hierarchical and complex designs, identification of the appropriate level for tests and full reporting of outcomes |
| ☒ | ☐ | Estimates of effect sizes (e.g. Cohen's *d*, Pearson's *r*), indicating how they were calculated |

*Our web collection on statistics for biologists contains articles on many of the points above.*

## Software and code

Policy information about availability of computer code

| Data collection | No software was used to collect data. |
|---|---|
| Data analysis | The source code is available under the MIT license at https://github.com/microsoft/hce-classification. |

For manuscripts utilizing custom algorithms or software that are central to the research but not yet described in published literature, software must be made available to editors and reviewers. We strongly encourage code deposition in a community repository (e.g. GitHub). See the Nature Portfolio guidelines for submitting code & software for further information.

## Data

Policy information about availability of data

All manuscripts must include a data availability statement. This statement should provide the following information, where applicable:

- Accession codes, unique identifiers, or web links for publicly available datasets
- A description of any restrictions on data availability
- For clinical datasets or third party data, please ensure that the statement adheres to our policy

The datasets used in this work were obtained from CELLxGENE (census version 2023-12-15). A preprocessed version of the data has been made available by the scTab study (Fischer et al. 2024): https://pklab.med.harvard.edu/felix/data/merlin_cxg_2023_05_15_sf-log1p.tar.gz. Ontology relationships were resolved using the Ontology Lookup Service (OLS): https://www.ebi.ac.uk/ols/ontologies/cl. Model checkpoints needed to reproduce the results in this work can be found at https://zenodo.org/records/17211022. Source data for Figures 1d, 1e, 2b, 2c, 2d and Supplementary Figures 1-4, 7, 9, 10 is available with this manuscript.

# Research involving human participants, their data, or biological material

Policy information about studies with human participants or human data. See also policy information about sex, gender (identity/presentation), and sexual orientation and race, ethnicity and racism.

| Reporting on sex and gender | N/A |
|---|---|
| Reporting on race, ethnicity, or other socially relevant groupings | N/A |
| Population characteristics | N/A |
| Recruitment | N/A |
| Ethics oversight | N/A |

Note that full information on the approval of the study protocol must also be provided in the manuscript.

# Field-specific reporting

Please select the one below that is the best fit for your research. If you are not sure, read the appropriate sections before making your selection.

☒ Life sciences  ☐ Behavioural & social sciences  ☐ Ecological, evolutionary & environmental sciences

For a reference copy of the document with all sections, see nature.com/documents/nr-reporting-summary-flat.pdf

# Life sciences study design

All studies must disclose on these points even when the disclosure is negative.

| Sample size | A formal sample size calculation was not conducted. Since we used publicly available datasets, the main authors of those corresponding papers determined the sample size. |
|---|---|
| Data exclusions | There was no exclusion of data. |
| Replication | Not applicable. We used publicly available datasets and did not have control over replication in the study design. |
| Randomization | Not applicable. We used publicly available datasets and did not have control over randomization in the study design. |
| Blinding | Not applicable. We used publicly available datasets and did not have control over blinding in the study design. |

# Reporting for specific materials, systems and methods

We require information from authors about some types of materials, experimental systems and methods used in many studies. Here, indicate whether each material, system or method listed is relevant to your study. If you are not sure if a list item applies to your research, read the appropriate section before selecting a response.

## Materials & experimental systems

| n/a | Involved in the study |
|---|---|
| ☒ | ☐ Antibodies |
| ☒ | ☐ Eukaryotic cell lines |
| ☒ | ☐ Palaeontology and archaeology |
| ☒ | ☐ Animals and other organisms |
| ☒ | ☐ Clinical data |
| ☒ | ☐ Dual use research of concern |
| ☒ | ☐ Plants |

## Methods

| n/a | Involved in the study |
|---|---|
| ☒ | ☐ ChIP-seq |
| ☒ | ☐ Flow cytometry |
| ☒ | ☐ MRI-based neuroimaging |

## Plants

Seed stocks

N/A

Novel plant genotypes

N/A

Authentication

N/A

