## [Peer Review File · Nature Computational Science]

Improving atlas-scale single-cell annotation models with hierarchical cross-entropy loss

Corresponding Author: Dr Lorin Crawford

Version 0:

Decision Letter:

**** Please ensure you delete the link to your author homepage in this e-mail if you wish to forward it to your co-authors. ****

Dear Dr Crawford,

Your manuscript "Hierarchical cross-entropy loss improves atlas-scale single-cell annotation models" has now been seen by 2 referees, whose comments are appended below. You will see that while they find your work of interest, they have raised points that need to be addressed before we can make a decision on publication.

The referees' reports seem to be quite clear. Naturally, we will need you to address **all** of the points raised.

While we ask you to address all of the points raised, the following points need to be substantially worked on: Please address the requests for additional comparisons to SOTA tools and additional details regarding the methodology.

Please use the following link to submit your revised manuscript and a point-by-point response to the referees' comments (which should be in a separate document to any cover letter):

Link Redacted

**** This url links to your confidential homepage and associated information about manuscripts you may have submitted or be reviewing for us. If you wish to forward this e-mail to co-authors, please delete this link to your homepage first. ****

To aid in the review process, we would appreciate it if you could also provide a copy of your manuscript files that indicates your revisions by making use of Track Changes or similar mark-up tools. Please also ensure that all correspondence is marked with your Nature Computational Science reference number in the subject line.

In addition, please make sure to upload a Word Document or LaTeX version of your text, to assist us in the editorial stage.

To improve transparency in authorship, we request that all authors identified as 'corresponding author' on published papers create and link their Open Researcher and Contributor Identifier (ORCID) with their account on the Manuscript Tracking System (MTS), prior to acceptance. ORCID helps the scientific community achieve unambiguous attribution of all scholarly contributions. You can create and link your ORCID from the home page of the MTS by clicking on 'Modify my Springer Nature account'. For more information please visit www.springernature.com/orcid.

We hope to receive your revised paper within three weeks. If you cannot send it within this time, please let us know.

I would also like to inform you in advance that moving forward, Ananya Rastogi (ananya.rastogi@nature.com) will serve as the handling editor for this manuscript. Please direct any future correspondence or inquiries regarding the submission to her.

Best regards,

Michelle Badri, PhD

Associate Editor, Research Cross-Journal Editorial Team for Nature Computational Science
Nature Portfolio

Reviewers comments:

Reviewer #1 (Remarks to the Author):

In this manuscript, the authors address the challenge of automated cell type annotation from single cell gene expression data (scRNAseq). They propose a loss function hierarchical cross-entropy (HCE), that incorporates the inherent hierarchical relationships between cell types. The authors design two analyses to demonstrate the performance improvements achieved with HCE compared to standard cross-entropy. First, they use the publicly available CELLxGENE census to define an In Distribution (ID) cell annotation task using donor-partitioned splits. In other words, for each dataset, some donors are included in the training dataset, and some are held-out the test dataset. Second, they use a later addition to CELLxGENE to define an Out of Distribution (OOD) task, where the test set is fully composed of datasets which are unseen during training (but which cover cell types that were seen during training). Using these tasks, the authors compare three models of increasing complexity to perform cell annotation. They demonstrated significant performances in cell annotation across model types and cell types.

I recommend this manuscript for publication due to its clear isolation of the effect of using an ontology-aware loss function in training cell type annotation models across several model architectures. The main contribution is defining the HCE, a novel extension of cross-entropy, and demonstrating that in otherwise-identical datasets and training settings and with identical model architectures, HCE leads to improvements in performance for most cell types. Whereas a standard cross-entropy loss treats all cell type labels as independent (implying that hierarchically-related cell types will compete for predictions), the HCE allows for sharing or accumulating information from related cell types to their parent nodes.

Cell annotation continues to be an important task in single-cell analyses with manual steps, and the field will benefit from having improved approaches for incorporating biological information about relationships between cell types. This work provides a nice argument that inductive bias in design of the loss function consistent with the hierarchical nature of cell types can offer substantial performance gains, offering an avenue for model improvement beyond simply increasing model size/complexity. The authors also provide a python implementation. My major comments involve contextualizing the work in the setting of other methods that leverage hierarchical cell type relationships in some way, adding detail to the methods regarding the Cell Ontology and HCE implementation, and improving the analysis and discussion to more intuitively understand performance and limitations of the HCE.

Major comments

- 1) This approach is not the first to leverage hierarchical relationships between cell types. Previously, OnClass, popSV, and even scTab have aimed to use hierarchical relationships between cell types, generally, and/or Cell Ontology, specifically, to improve cell type annotations. Could the authors briefly situate their strategy (ontology-aware loss function) relative to these others (e.g. computation of a new dimensionality reduction based on an input ontology) that incorporate cell hierarchy/ontological relationships. This could be done in the introductory parts the main text, in the section on automated cell annotation methods, or in the discussion at the end.
- 2) Generally, the Methods could benefit from much more details regarding the provenance of the input cell ontology, its representation and associated analyses. In particular, define connected and isolated nodes as used in Supplementary Figure 3, in the context of the Cell Ontology DAG. The source of the ontology should be briefly mentioned in the main text, and then described in more detail in the Methods.
- 3) The authors provide details about the HCE implementation via matrix multiplication between the class probabilities and a reachability matrix defining the DAG, and these would be nice details to add to the Methods.
- 4) The performance improvements are very clear, but I think the work could benefit from greater understanding of the mis-classifications, both hidden and measured:
 - a) The authors state that the “predicted label is considered correct if it exactly matches the ground-truth label or if it corresponds to a descendant of the ground-truth label in the ontology graph”. In the case of a granular cell types seen in the training dataset with a broader label in the test set, use of the HCE enables correction of the labels through propagation of the probabilities. While HCE correctly assigns a shared parent label in cases of uncertainty, does this come at the cost of decreased precision between closely related sibling types? For example, are there instances where HCE is more likely than standard cross-entropy to confuse an ' α - β T cell' with a ' γ - δ T cell' because they share the parent 'T cell' label and would equally be reported as correct?
 - b) The case of unseen cell types represents a more extreme OOD scenario. The authors describe a “fallback scenario” where unseen cell types are assigned a reasonable broader label rather than an inaccurate granular label. Could the authors alter the cell types included in their training/test datasets to explore this, and comment on the performance of HCE in the case of unseen cell types (given that this is a common scenario in single-cell atlas studies?) This would more directly address the fallback scenario briefly described in the text. Since such cases would be labelled as incorrect, it could also be interesting to return a separate vector for each cell, indicating whether it was classified as a parent/ancestor of the true label, or not.
- 5) The utility of the HCE depends on both having a good ontology capturing hierarchical cell type relationships (such as the

Cell Ontology), as well as having datasets with labels that match terminology of nodes in the ontology. This is a practical challenge to using the HCE in single cell annotation settings. Could the authors address this point in their discussion? What is the state of existing cell atlases in terms of having harmonized labels aligned to cell ontologies?

Minor comments

- 6) The “adjusted probabilities”, following accumulation of probabilities for each node across its full set of descendants, are no longer true probabilities which sum to 1 across the full set of labels. Might it be more appropriate to refer to these more generally as an “adjusted score”?
- 7) Figure 1e: indicate on the figure or legend that dashed bars represent the In-distribution performance for each model
- 8) In barplots with per cell type F1 metrics (e.g. Figure 2c), clarify in the legend what individual points correspond to
- 9) Figure 2d: does this represent the DAG for the entire Cell Ontology used in the study?
- 10) Supplementary Figure 2: specify the base of the log scale for the x-axis
- 11) Supplementary Figures 6-8: provide legends for the node size and colors

Statistical tests used in the manuscript appear reasonable. Authors compare significance of changes in cell type annotation with cross-entropy vs HCE loss for each model using paired t-tests followed by multiple testing correction.

Reviewer #1 (Remarks on code availability):

The authors have provided a thorough codebase which documents the principles for computing HCE via a reachability matrix, and provided a python implementation of the loss function, given model output logits, the true class labels, and the reachability matrix associated with the ontology.

While the python HCE implementation example in the README is appreciated, it was difficult to understand the relationship between the present study, scTab, and the cellnet models, and to track down the HCE loss function used in the analyses presented in the paper to the authors' updated cellnet implementation. This could be resolved with improved documentation of the structure of the codebase, and a brief guide for how to use the provided scripts to reproduce the experiments and evaluation figures shown in the manuscript.

Reviewer #2 (Remarks to the Author):

This paper presents a strategy for improving cell type assignments using the hierarchical nature of cell types. This is a fundamental problem in analyzing single-cell and spatial genomic data, with good potential for broad practical value if significant improvements can be achieved on the prior art. The method proposed here is a modification of standard methods that enforces consistency between different resolutions of cell type assignment, as a way of dealing better with inconsistencies in levels of annotations between data sets by taking advantage of their inherently hierarchical organization. It is a good idea, although not obviously a big advance on prior art. The idea is tested with three kinds of classifiers, showing generally a very good improvement in accuracy of the assignments for all three classifiers. It does a good job of looking into which kinds of assignments tend to benefit or not from this extension. Overall, it looks like a valuable, if not dramatic, practical advance on an important problem. Nonetheless, there are a few points I would like to ask about:

1. The paper uses three standard models to test the improvement achieved by hierarchical cross-entropy model. While they seem a reasonable and technically diverse set, I would want to see a stronger justification for why specifically these three and not any of the many other methods that have been or could be applied to this problem.
2. Related to that prior point, I feel it is important to include some comparison to state of the art methods for the problem. I realize the point of this paper is to test the value added by this one idea and not create a single best piece of software. However, I believe it is important to establish whether adding the hierarchical method to standard classifiers leads to something that is at least close to competitive with popular state of the art models, either with or without the hierarchical cross entropy.
3. I believe the scoring scheme for the validation testing could use some further justification. Essentially, the scheme counts an assignment as correct if it is a descendent of the assignment recorded as the ground truth in the test data. While this seems a fair policy given limits of the data, it does have the potential to score as correct some answers that are genuinely incorrect and I would wonder if it is possible to handle that better. How results would the results compare, for example, if you exclude cases where the prediction and ground truth are at different levels of resolution?
4. It would be useful to see a statement of how model training and hyperparameter selection were done. Were hyperparameters tuned over the course of the study or, if not, is there a justification for the values chosen? If they were tuned, how was that done? I believe the details of this are important to be confident that hyperparameters were not unfairly tuned to the test.
5. I could also use more convincing about how out-of-distribution data was handled in establishing testing and training data. While superficially the procedure chosen --- splitting data points based on when they were added to the database --- seems fair, there would seem to be a risk of data not being fully independent between training and validation. I would want to see

more evidence that there is not unreasonable redundancy between the data sets or unfair biases that could lead to training on batch effects (e.g., skew in which labs are generating which kinds of data). Ideally, we would want to see that the method is effective when trained on one data set and tested on a fully independent dataset.

Reviewer #2 (Remarks on code availability):

I have looked through the code but not installed or run it as I do not have access to suitable compute resources to duplicate the results in the paper. The code appears well organized and written and suitably documented, but I cannot comment on ease of installation/use or whether it runs as expected.

Version 1:

Decision Letter:

Our ref: NATCOMPUTSCI-25-1688A

22nd October 2025

Dear Dr. Crawford,

Thank you for submitting your revised manuscript "Hierarchical cross-entropy loss improves atlas-scale single-cell annotation models" (NATCOMPUTSCI-25-1688A). It has now been seen by the original referees and their comments are below. The reviewers find that the paper has improved in revision, and therefore we'll be happy in principle to publish it in Nature Computational Science, pending minor revisions to satisfy the referees' final requests and to comply with our editorial and formatting guidelines.

TRANSPARENT PEER REVIEW

Nature Computational Science offers a transparent peer review option for original research manuscripts. We encourage increased transparency in peer review by publishing the reviewer comments, author rebuttal letters and editorial decision letters if the authors agree. Such peer review material is made available as a supplementary peer review file. **Please remember to choose, using the manuscript system, whether or not you want to participate in transparent peer review.**

Thank you again for your interest in Nature Computational Science. Please do not hesitate to contact me if you have any questions.

Sincerely,

Kaitlin McCardle, PhD
Senior Editor
Nature Computational Science

ORCID

Author names using non-Roman characters

Nature Portfolio journals can support presentation of author names using non-Roman characters in the HTML version of the article. If you wish to, please include author names in parentheses after the Roman-character spelling; [see example online here](https://www.nature.com/articles/s44222-024-00258-2). Currently supported scripts are: Arabic, Chinese, Cyrillic, Devanagari, Greek, Hebrew, Hangul, Japanese and Persian. You will be asked to verify the rendering is correct at proof stage.

Reviewer #1 (Remarks to the Author):

The authors have addressed all my concerns.

Reviewer #2 (Remarks to the Author):

The responses to the prior reviews largely address my previous concerns. The approach and scope of work are now better explained and justified. The response provides some additional assurance that the testing is handled fairly. I do not have any new critiques to add and accept that the prior ones have been addressed as well as can reasonably be expected. However, some of my original concern still stands regarding the possibility that the testing could be training in part on batch effects and that more might be done to exclude this possibility by testing on fully independent testing data. My initial impression still stands that this is a useful advance for an important problem, although not a dramatic one conceptually or practically, relative to the prior art.

Reviewer #2 (Remarks on code availability):

I did not have any critiques about the code previously and so did not do any new review of it. However, I have looked through the updated documentation in the Github provided in response to reviewer 1's critiques and agree it is nicely done and a valuable addition.

Version 2:

Decision Letter:

Dear Dr Crawford,

We are pleased to inform you that your Brief Communication "Improving atlas-scale single-cell annotation models with hierarchical cross-entropy loss" has now been accepted for publication in Nature Computational Science.

Once your manuscript is typeset, you will receive an email with a link to choose the appropriate publishing options for your paper and our Author Services team will be in touch regarding any additional information that may be required.

Authors may need to take specific actions to achieve compliance with funder and institutional open access mandates. If your research is supported by a funder that requires immediate open access (e.g. according to [Plan S principles](https://www.springernature.com/gp/open-science/plan-s-compliance) or the [NIH public access policy](https://www.springernature.com/gp/open-science/us-federal-agency-compliance)) then you should select the gold OA route, and we will direct you to the compliant route where possible. Because authors warrant under our subscription licensing terms that they haven't committed to licensing any version of their article under a licence inconsistent with the terms of our agreement – including the applicable embargo period – publication under the subscription model isn't suitable for authors whose funders require no embargo.

Acceptance of your manuscript is conditional on all authors' agreement with our publication policies (see <https://www.nature.com/natcomputsci/for-authors>). In particular your manuscript must not be published elsewhere and there must be no announcement of the work to any media outlet until the publication date (the day on which it is uploaded onto our web site).

Before your manuscript is typeset, we will edit the text to ensure it is intelligible to our wide readership and conforms to house style. We look particularly carefully at the titles of all papers to ensure that they are relatively brief and understandable.

Once your manuscript is typeset, you will receive a link to your electronic proof via email with a request to make any corrections within 48 hours. If, when you receive your proof, you cannot meet this deadline, please inform us at rjsproduction@springernature.com immediately.

If you have queries at any point during the production process then please contact the production team at rjsproduction@springernature.com.

We welcome the submission of potential cover material (including a short caption of around 40 words) related to your

manuscript; suggestions should be sent to Nature Computational Science as electronic files (the image should be 300 dpi at 210 x 297 mm in either TIFF or JPEG format). We also welcome suggestions for the Hero Image, which appears at the top of our [home page](http://www.nature.com/natcomputsci); these should be 72 dpi at 1400 x 400 pixels in JPEG format. Please note that such pictures should be selected more for their aesthetic appeal than for their scientific content, and that colour images work better than black and white or grayscale images. Please do not try to design a cover with the Nature Computational Science logo etc., and please do not submit composites of images related to your work. I am sure you will understand that we cannot make any promise as to whether any of your suggestions might be selected for the cover of the journal.

Best regards,

Kaitlin McCardle, PhD
Senior Editor
Nature Computational Science

P.S. Click on the following link if you would like to recommend Nature Computational Science to your librarian: <https://www.springernature.com/gp/librarians/recommend-to-your-library>

** Visit the Springer Nature Editorial and Publishing website at <http://editorial-jobs.springernature.com> for more information about our career opportunities. If you have any questions please click [here](mailto:editorial.publishing.jobs@springernature.com). **

We thank the reviewers for their comments and suggestions. We appreciate the time and effort taken to evaluate our work and are grateful for the suggestions that have helped us to improve our submission.

Overview of changes

We have carefully considered each point raised and have addressed them thoroughly in the revised manuscript. Following the helpful feedback from reviewers, we have now **(1)** expand the main text to include a more thorough review of existing approaches that leverage hierarchical relationships between cell types; **(2)** provide more explicit details about how we implement the hierarchical cross entropy loss in practice; and **(3)** thoroughly explain how model training and hyperparameter selection is done in our analyses and evaluations.

We copied the reviewers' comments in full and listed them in blue. Below, we respond to each comment in detail. Direct quotes from our revision are given in ***bold italics***. We are very grateful for the constructive criticism and do agree that addressing these concerns has improved our manuscript.

Reviewer #1 (Remarks to the Author):

In this manuscript, the authors address the challenge of automated cell type annotation from single cell gene expression data (scRNAseq). They propose a loss function hierarchical cross-entropy (HCE), that incorporates the inherent hierarchical relationships between cell types. The authors design two analyses to demonstrate the performance improvements achieved with HCE compared to standard cross-entropy. First, they use the publicly available CELLxGENE census to define an In Distribution (ID) cell annotation task using donor-partitioned splits. In other words, for each dataset, some donors are included in the training dataset, and some are held-out the test dataset. Second, they use a later addition to CELLxGENE to define an Out of Distribution (OOD) task, where the test set is fully composed of datasets which are unseen during training (but which cover cell types that were seen during training). Using these tasks, the authors compare three models of increasing complexity to perform cell annotation. They demonstrated significant performances in cell annotation across model types and cell types.

I recommend this manuscript for publication due to its clear isolation of the effect of using an ontology-aware loss function in training cell type annotation models across several model architectures. The main contribution is defining the HCE, a novel extension of cross-entropy, and demonstrating that in otherwise-identical datasets and training settings and with identical model architectures, HCE leads to improvements in performance for most cell types. Whereas a standard cross-entropy loss treats all cell type labels as independent (implying that hierarchically related cell types will compete for predictions), the HCE allows for sharing or accumulating information from related cell types to their parent nodes.

Cell annotation continues to be an important task in single-cell analyses with manual

steps, and the field will benefit from having improved approaches for incorporating biological information about relationships between cell types. This work provides a nice argument that inductive bias in design of the loss function consistent with the hierarchical nature of cell types can offer substantial performance gains, offering an avenue for model improvement beyond simply increasing model size/complexity. The authors also provide a python implementation. My major comments involve contextualizing the work in the setting of other methods that leverage hierarchical cell type relationships in some way, adding detail to the methods regarding the Cell Ontology and HCE implementation, and improving the analysis and discussion to more intuitively understand performance and limitations of the HCE.

We thank reviewer for the thoughtful assessment and supportive comments about our initial submission. The issues raised in this summary are further detailed in the reviewer's major/minor comments below and we address each of them individually.

Major Comments:

This approach is not the first to leverage hierarchical relationships between cell types. Previously, OnClass, popSV, and even scTab have aimed to use hierarchical relationships between cell types, generally, and/or Cell Ontology, specifically, to improve cell type annotations. Could the authors briefly situate their strategy (ontology-aware loss function) relative to these others (e.g. computation of a new dimensionality reduction based on an input ontology) that incorporate cell hierarchy/ontological relationships. This could be done in the introductory parts the main text, in the section on automated cell annotation methods, or in the discussion at the end.

This is a great point, and we appreciate the reviewer for making this suggestion to further contextualize our work for readers. In the revised manuscript, we now include the following paragraph in the main text (see new lines 71-88):

While there are methods that leverage ontological information for cell type annotation, they do not enforce hierarchical consistency as an integral part of their predictive framework. For example, OnClass maps both transcriptomic profiles and cell ontology structure into a joint embedding space, enabling both the annotation of unseen cell types and the identification of marker genes (Wang et al. 2019, Nature Communications). However, it operates primarily as a nearest-neighbor or embedding search algorithm and does not couple hierarchical relationships to the learned probabilities for each cell. As a result, sibling classes or intermediate states can still be misassigned if their embeddings overlap in feature space. As another example, popV aggregates predictions from multiple classifiers using ontology-based voting, producing robust consensus labels and uncertainty estimates for ambiguous or outlier populations (Ergen et al. 2024, Nature Genetics). Yet, the ontology is used only as a scaffold for post hoc reconciliation and not as a guide for model optimization. This means that hierarchical constraints are not

encoded in training and possible conflicts or inconsistencies in the ensemble are resolved heuristically. In contrast, SCimilarity focuses on metric learning for scalable, cross-study retrieval of transcriptionally similar cells, using the ontology at training time to exclude ambiguous annotation pairs when sampling triplets for a contrastive loss function (Heimberg et al. 2024, Nature). The learned representation supports high-quality search and transfer tasks but is not directly optimized for hierarchical or taxonomic consistency when determining class probabilities. In summary, unlike these approaches, our hierarchical cross-entropy loss explicitly encodes hierarchical dependencies and relationships into the model's objective, ensuring all predictions respect the structure of the cell ontology.

Generally, the Methods could benefit from much more details regarding the provenance of the input cell ontology, its representation and associated analyses. In particular, define connected and isolated nodes as used in Supplementary Figure 3, in the context of the Cell Ontology DAG. The source of the ontology should be briefly mentioned in the main text, and then described in more detail in the Methods.

We appreciate the reviewer's feedback, and we apologize for this lack of detail in our initial submission. In revision, we have added a new subsection to the Methods entitled "Cell ontology" (see new lines 144-160). It reads:

We used the cell ontology obtained from the Ontology Lookup Service (OLS) at EMBL-EBI as the hierarchical scaffold for all analyses (Jupp et al. 2015, Workshop on Semantic Web Applications and Tools for Life Sciences). The ontology was represented as a directed acyclic graph (DAG), where nodes correspond to cell types and directed edges correspond to *is_a* subtype relationships. We restricted the ontology to the 164 distinct cell types observed in the training set (see "Training and evaluation datasets" in the previous subsection). In CELLxGENE, which is the atlas used in our study, cell types are annotated by the original data contributors and then harmonized by mapping each label to the closest cell ontology term as specified by the portal's data schema. While the cell ontology offers a valuable scaffold for representing hierarchical relationships among cell types, it is important to note that its structure is continuously being revised where certain definitions and mappings between cell types remain under active refinement.

Because each cell type corresponds to a node in the DAG, we can further classify them based on the type of node they represent. A node was defined as a leaf if it had no children in the pruned ontology and as an internal node if it had at least one child. We also distinguished between connected nodes, which had at least one parent or child present in the curated training set, and isolated nodes, which had none of their ancestors or descendants represented in the training data. These definitions were used to assess how the hierarchical loss propagates information across the ontology (e.g., Supplementary Fig. 3).

The authors provide details about the HCE implementation via matrix multiplication

between the class probabilities and a reachability matrix defining the DAG, and these would be nice details to add to the Methods.

We agree with the reviewer and thank them for their suggestion. In the revised manuscript, we have added a new subsection to the Methods entitled “Implementation details for the HCE loss” (see new lines 221-241). It reads:

We implemented the hierarchical cross-entropy loss using a reachability matrix $R \in \{0, 1\}^{C \times C}$ where element $R_{ij} = 1$ if the j -th class is reachable from the i -th class (meaning j is either i itself or j is a descendant of i in the hierarchy), and $R_{ij} = 0$ otherwise. The reachability relation encoded in this matrix is a partial order and has the following mathematical properties:

- **reflexive: every class is reachable from itself (diagonal elements are 1);**
- **antisymmetric: if class i can reach j and j can reach i , then $i = j$;**
- **transitive: if class i can reach j and j can reach k , then i can reach k .**

Indeed, the reachability matrix represents the transitive closure of the inverted adjacency matrix of the hierarchical DAG structure. Since the original DAG encodes i s_a relationships from child to parent, we invert the edge directions to enable parent-to-descendant reachability, ensuring reflexivity by setting the diagonal to 1. Each trained model outputs a raw probability distribution $p = (p_1, \dots, p_C)$ over the class labels. The adjusted scores are computed via matrix-vector multiplication: $s = Rp$, which efficiently aggregates descendant probabilities for each class. We then apply a log transformation with numerical stability $\log(s + \epsilon)$, where $\epsilon = 10^{-6}$. The final loss uses a weighted negative log-likelihood as implemented in PyTorch, with class weights computed following scikit-learn's `compute_class_weight` approach: $w_i = N / (C \cdot n_i)$, where N is the total number of samples, C is the number of classes, and n_i is the count of samples for the class i . The complete loss for a single training sample x with true label t is

$$\mathcal{L}_{HCE}(x) = -w_t \log(s_t + \epsilon).$$

This formulation maintains consistency with the models trained with the weighted cross-entropy, while incorporating hierarchical structure through efficient matrix operations.

The performance improvements are very clear, but I think the work could benefit from greater understanding of the mis-classifications, both hidden and measured:

- a) The authors state that the “predicted label is considered correct if it exactly matches the ground-truth label or if it corresponds to a descendant of the ground-truth label in the ontology graph”. In the case of a granular cell types seen in the training dataset with a broader label in the test set, use of the HCE enables correction of the labels through propagation of the probabilities. While HCE

correctly assigns a shared parent label in cases of uncertainty, does this come at the cost of decreased precision between closely related sibling types? For example, are there instances where HCE is more likely than standard cross-entropy to confuse an ' α - β T cell' with a ' γ - δ T cell' because they share the parent 'T cell' label and would equally be reported as correct?

We thank the reviewer for raising this important point. In full transparency, we spent a lot of time thinking through these different scenarios. Across all architectures, we did not observe systematic decreases in precision between sibling types. In fact, the performances for both the MLP model and TabNet with the HCE were statistically significant improvements over their respective baselines using the standard cross-entropy. For example, in both Figure 2c and Supplementary Figure 4, many of these instances included siblings such as CD4+ α - β T cells and CD8+ α - β T cells.

One important thing that we should clarify is that, while HCE propagates probability mass from a subtype to its parent, it does not transfer probability between siblings. The model must still distinguish α - β T cells from γ - δ T cells, and probability assigned to one does not increase the adjusted score of the other.

- b) The case of unseen cell types represents a more extreme OOD scenario. The authors describe a “fallback scenario” where unseen cell types are assigned a reasonable broader label rather than an inaccurate granular label. Could the authors alter the cell types included in their training/test datasets to explore this, and comment on the performance of HCE in the case of unseen cell types (given that this is a common scenario in single-cell atlas studies?) This would more directly address the fallback scenario briefly described in the text. Since such cases would be labelled as incorrect, it could also be interesting to return a separate vector for each cell, indicating whether it was classified as a parent/ancestor of the true label, or not.

This is another great point raised by the reviewer, and we apologize for the confusion. Our work, like scTab (Fischer et al. 2024, *Nature Communications*), uses a closed-world label set: the 164 cell ontology terms seen in training define the prediction space, and all test labels are drawn from this set. In the revision, we have revised the text to make this clearer by avoiding the phrase that suggested we were trying to predict unseen-labels (see revised lines 195-220).

The original wording from our initial submission stemmed from the fact that, given the upward-propagation mechanism, HCE could in principle be implemented as a probability distribution defined only over leaf nodes. Instead, we deliberately maintain a raw probability distribution over all nodes in the hierarchy, including parent nodes. This design allows the model being trained to assign non-zero probability mass to broader labels; for example, when it is less confident (see plot below). While argmax selection typically yields leaf-node predictions (which is optimal during training, as these benefit most from upward

propagation), we still observe non-zero probabilities on internal nodes in practice. In a small number of cases, a non-leaf node attains sufficiently high raw probability to be selected as the predicted cell type. For instance, dendritic cells and γ - δ T cells are both internal nodes and are predicted ~600 and ~300 times, respectively. The other plot below shows the total count of non-leaf (internal node) predictions.

The utility of the HCE depends on both having a good ontology capturing hierarchical cell type relationships (such as the Cell Ontology), as well as having datasets with labels that match terminology of nodes in the ontology. This is a practical challenge to using the HCE in single cell annotation settings. Could the authors address this point in their discussion? What is the state of existing cell atlases in terms of having harmonized labels aligned to cell ontologies?

These are great points, and we appreciate the suggestion. First, in the revision, we have now added the following sentence to the Discussion (120-122):

It is important to note that HCE relies on a predefined, labeled DAG, and while the cell ontology serves as a valuable reference, it is continuously evolving, with ongoing updates to cell type definitions and their hierarchical relationships.

Second, regarding the state of existing cell atlases in terms of having harmonized labels aligned to cell ontologies, we have added this sentence in the new “Cell ontology subsection within the Methods (see new lines 149-151):

In CELLxGENE, which is the atlas used in our study, cell types are annotated by the original data contributors and then harmonized by mapping each label to the closest cell ontology term as specified by the portal’s data schema.

Minor Comments:

The “adjusted probabilities”, following accumulation of probabilities for each node across its full set of descendants, are no longer true probabilities which sum to 1 across the full set of labels. Might it be more appropriate to refer to these more generally as an “adjusted score”?

This is a great point, and we appreciate the reviewer for bringing this up. In the revision we now use the phrase “adjusted score” and have updated the corresponding notation (see new lines 194-241).

Figure 1e: indicate on the figure or legend that dashed bars represent the In-distribution performance for each model.

Yes. We have added the following sentence to the caption: ***The dashed red bars indicate the in-distribution performances for comparison.***

In barplots with per cell type F1 metrics (e.g. Figure 2c), clarify in the legend what individual points correspond to.

Great catch. We have added the following sentence to the captions of Figure 2 and Supplementary Figures 1, 4, 9, and 10: ***Each dot represents the performance of an individual run (color coding remains the same as in the legend).***

Figure 2d: does this represent the DAG for the entire Cell Ontology used in the study?

Yes, the DAG in this figure has 164 nodes corresponding to all cell types seen in the training set. We have added the following clarification (or similar) to the captions of Figure 2 and Supplementary Figures 6-8: **Note that this DAG consists of all 164 cell types seen in the training set.**

Supplementary Figure 2: specify the base of the log scale for the x-axis.

We thank the reviewer for making this suggestion. We want to note that any logarithm base is equivalent up to a rescaling, so the base mainly affects which tick marks are shown in the figure. We used the default (base 10) which we now clarify in the caption. Because the data in this plot does not span many orders of magnitude, the plotting library rendered the tick marks it deemed most appropriate.

Supplementary Figures 6-8: provide legends for the node size and colors.

We thank the reviewer for this comment. We have now added the same legend as in Figure 2d to Supplementary Figures 6-8.

Statistical tests used in the manuscript appear reasonable. Authors compare significance of changes in cell type annotation with cross-entropy vs HCE loss for each model using paired t-tests followed by multiple testing correction.

We appreciate the reviewer for being so thorough while evaluating our work.

Reviewer #1 (Remarks on code availability):

The authors have provided a thorough codebase which documents the principles for computing HCE via a reachability matrix, and provided a python implementation of the loss function, given model output logits, the true class labels, and the reachability matrix associated with the ontology.

While the python HCE implementation example in the README is appreciated, it was difficult to understand the relationship between the present study, scTab, and the cellnet models, and to track down the HCE loss function used in the analyses presented in the paper to the authors' updated cellnet implementation. This could be resolved with improved documentation of the structure of the codebase, and a brief guide for how to use the provided scripts to reproduce the experiments and evaluation figures shown in the manuscript.

We thank the reviewer for this excellent suggestion. Following these comments, we have now completely updated the GitHub repo with more documentation to both facilitate the implementation of the HCE methodology and the reproducibility of our analyses. Please see <https://github.com/microsoft/hce-classification> for these changes (and lines 253-254).

Reviewer #2 (Remarks to the Author):

This paper presents a strategy for improving cell type assignments using the hierarchical nature of cell types. This is a fundamental problem in analyzing single-cell and spatial genomic data, with good potential for broad practical value if significant improvements can be achieved on the prior art. The method proposed here is a modification of standard methods that enforces consistency between different resolutions of cell type assignment, as a way of dealing better with inconsistencies in levels of annotations between data sets by taking advantage of their inherently hierarchical organization. It is a good idea, although not obviously a big advance on prior art. The idea is tested with three kinds of classifiers, showing generally a very good improvement in accuracy of the assignments for all three classifiers. It does a good job of looking into which kinds of assignments tend to benefit or not from this extension. Overall, it looks like a valuable, if not dramatic, practical advance on an important problem. Nonetheless, there are a few points I would like to ask about.

We appreciate the reviewer's feedback and constructive criticism. We have taken each of the comments and have incorporated relevant changes in the revised version of the manuscript. We believe that these additional analyses and discussion have greatly improved the paper. Our point-by-point response is provided below.

Major Comments:

The paper uses three standard models to test the improvement achieved by hierarchical cross-entropy model. While they seem a reasonable and technically diverse set, I would want to see a stronger justification for why specifically these three and not any of the many other methods that have been or could be applied to this problem.

We appreciate the reviewer for bringing up this important point. First, as the reviewer points out, the selection of these three models was as an experimental design choice which was meant to demonstrate the relative improvements gained by using the hierarchical cross-entropy is architecture-agnostic. In part, these models were specifically chosen to maintain a direct parallel with the original scTab study (Fischer et al. 2024, *Nature Communications*). The authors of that work provide a comprehensive benchmark spanning model frameworks with varying architectural complexity. The linear model, multilayer perceptron, and TabNet were the three end-to-end models evaluated. Importantly, each of these models had training code available and we could ensure that each were exposed to the same training datasets. We believe that the inclusion of pretrained models (where we would instead evaluate fine-tuning using cross-entropy versus hierarchical cross-entropy) would create confounding effects on the true efficacy of the HCE approach.

Related to that prior point, I feel it is important to include some comparison to state-of-the-art methods for the problem. I realize the point of this paper is to test the value added by this one idea and not create a single best piece of software. However, I believe it is

important to establish whether adding the hierarchical method to standard classifiers leads to something that is at least close to competitive with popular state of the art models, either with or without the hierarchical cross entropy.

We appreciate the reviewer for making this thoughtful suggestion. First, the reviewer is correct: the main goal of our evaluations was not to re-run comprehensive benchmarks to identify the best model architectures, but instead to isolate the incremental value of the loss function. To that end, we keep model settings fixed and show that replacing cross-entropy with the hierarchical cross-entropy yields consistent gains in out-of-distribution analyses. We make this clearer now with the following addition to the text in the Methods (see new lines 184-189):

The models using cross-entropy (CE) versus hierarchical cross-entropy (HCE) share identical architecture and hyperparameter settings; the loss term is the only change that is different between them. Specifically, for the models with CE, we used the best hyperparameters available according to the original scTab study. For the models using the HCE loss, we did not perform additional hyperparameter tuning and instead kept the (possibly suboptimal) hyperparameters used for the models with CE.

Second, in our work, the TabNet model from the scTab study was meant to serve as the state-of-the-art model architecture. In Fischer et al. (2024), it was shown that TabNet outperformed both scGPT (Cui et al. 2024, *Nature Methods*) and the universal cell embedding (UCE) model (Rosen et al., *bioRxiv*) in cell type annotation, regardless of whether those algorithms were used zero-shot or fine-tuned. To highlight this, we included a copy of Supplementary Figure 1 from Fischer et al. (2024) below.

Supp. Figure 1: Comparison of scTab performance versus single-cell foundation models (scGPT and UCE).

Third, we want to gently push back on the idea that more complicated models should be held as the gold standard for the cell type annotation task. In work that was conducted by a subset of our co-authors, the zero-shot capabilities of scGPT and Geneformer to separate known cell types across multiple datasets was evaluated (see Figure 1 from Kedzierska et al. 2025, *Genome Biology*). There it was shown that both models perform worse than simply selecting highly variable genes (HVG) and using more established methods such as Harmony (Korsunsky et al. 2019, *Nature Methods*) and scVI (Lopez et al. 2018, *Nature Methods*) --- the latter of which is an example of a simple multilayer perceptron. Importantly, these results have also been shown by other groups. For example, Boiarsky et al. (2024) demonstrated that a simple logistic regression baseline outperforms or performs comparably to methods like scBERT (Yang et al. 2022, *Nature Machine Intelligence*) on the fine-tuning task of cell-type annotation. Similar stories can be found in other recent studies (e.g., Atti and Subramaniam, *bioRxiv*; Liu et al., *bioRxiv*).

Lastly, as we mentioned in the previous comment, we needed to use models that had training code available. This was so that we could ensure that each method was exposed to the same training and test dataset splits.

I believe the scoring scheme for the validation testing could use some further justification. Essentially, the scheme counts an assignment as correct if it is a descendent of the assignment recorded as the ground truth in the test data. While this seems a fair policy given limits of the data, it does have the potential to score as correct some answers that are genuinely incorrect, and I would wonder if it is possible to handle that better. How results would the results compare, for example, if you exclude cases where the prediction and ground truth are at different levels of resolution?

This is important to clarify, and we appreciate the reviewer for raising this point. As shown in Figure 2c and Supplementary Figure 4, models using the HCE approach had statistically significant improvements over their respective baselines using the standard cross-entropy (except for a couple cell types when using the linear model, which may be due to the limited complexity of linear models). Even when restricting evaluation to leaves, where the only correct predictions are exact matches (i.e., at the same level of resolution), we still see similar results in terms of relative improvements (see Supplementary Figure 3b). Moreover, it is worth noting that all models essentially always predict leaf nodes (99.6% of the predictions) as this is beneficial to multiple nodes in the hierarchy.

It would be useful to see a statement of how model training and hyperparameter selection were done. Were hyperparameters tuned over the course of the study or, if not, is there a justification for the values chosen? If they were tuned, how was that done? I believe the details of this are important to be confident that hyperparameters were not unfairly tuned to the test.

This is a great suggestion. In the revised version of the manuscript, we added the following text to the “Model details” subsection in the Methods (see new lines 184-189):

The models using cross-entropy (CE) versus hierarchical cross-entropy (HCE) share identical architecture and hyperparameter settings; the loss term is the only change that is different between them. Specifically, for the models with CE, we used the best hyperparameters available according to the original scTab study. For the models using the HCE loss, we did not perform additional hyperparameter tuning and instead kept the (possibly suboptimal) hyperparameters used for the models with CE.

I could also use more convincing about how out-of-distribution data was handled in establishing testing and training data. While superficially the procedure chosen --- splitting data points based on when they were added to the database --- seems fair, there would seem to be a risk of data not being fully independent between training and validation. I would want to see more evidence that there is not unreasonable redundancy between the data sets or unfair biases that could lead to training on batch effects (e.g., skew in which labs are generating which kinds of data). Ideally, we would want to see that the method is effective when trained on one data set and tested on a fully independent dataset.

We appreciate the reviewer's concern regarding out-of-distribution (OOD) testing. This is important to clarify. First, across all models, using the HCE loss leads to improvements on the 21 newly added studies collected between May and December 2023 (with the sole exception of one study for the linear model, Supplementary Fig. 1), supporting its robustness across diverse studies.

Unfortunately, CELLxGene does not provide sufficient metadata to systematically identify the contributing labs or experimental protocols to investigate contributions from different types of batch effects. Still, we sought to address this issue by highlighting cases where OOD studies introduce entirely new tissues or disease contexts absent from the training set. In these settings, HCE consistently outperforms CE, showing that the observed gains are not limited to matched distributions but extend to genuine context shifts (see new Supplementary Figures 9 and 10, also copied below for convenience). To make this explicit, we added the following clarification to the revised manuscript (see new lines 111-113):

Finally, these gains extend to cells observed in new contexts, including across diseases and tissues not seen in the training set where we also observe consistent improvements (Supplementary Figs. 9 and 10).

Reviewer #2 (Remarks on code availability):

I have looked through the code but not installed or run it as I do not have access to suitable compute resources to duplicate the results in the paper. The code appears well organized and written and suitably documented, but I cannot comment on ease of installation/use or whether it runs as expected.

We thank the reviewer for the thoughtful and thorough review of our work.

Supplementary Figure 9. Performance gains from the hierarchical cross-entropy (HCE) loss for different diseases for the linear classifier, multilayer perceptron (MLP), and TabNet. Improvements are measured relative to the same models trained with standard cross-entropy loss. Highlighted in pink are novel diseases in the test set that were not seen in the training set. Each dot represents the performance of an individual run (color coding remains the same as in the legend).

Supplementary Figure 10. Performance gains from the hierarchical cross-entropy (HCE) loss for different tissues for the linear classifier, multilayer perceptron (MLP), and TabNet. Improvements are measured relative to the same models trained with standard cross-entropy loss. Highlighted in pink are novel tissues in the test set that were not seen in the training set. Each dot represents the performance of an individual run (color coding remains the same as in the legend).

We thank the reviewers for their comments and suggestions. We appreciate the time and effort taken to evaluate our work and are grateful for the suggestions that have helped us to improve our submission.

Overview of changes

We have carefully considered the remaining points raised and have addressed them thoroughly in the revised manuscript.

We copied the reviewers' comments in full and listed them in blue. Below, we respond to each comment in detail. Direct quotes from our revision are given in ***bold italics***. We are very grateful for the constructive criticism and do agree that addressing these concerns has improved our manuscript.

Reviewer #1 (Remarks to the Author):

The authors have addressed all my concerns.

Reviewer #2 (Remarks to the Author):

The responses to the prior reviews largely address my previous concerns. The approach and scope of work are now better explained and justified. The response provides some additional assurance that the testing is handled fairly. I do not have any new critiques to add and accept that the prior ones have been addressed as well as can reasonably be expected. However, some of my original concern still stands regarding the possibility that the testing could be training in part on batch effects and that more might be done to exclude this possibility by testing on fully independent testing data. My initial impression still stands that this is a useful advance for an important problem, although not a dramatic one conceptually or practically, relative to the prior art.

We appreciate the reviewer's continued feedback and constructive criticism. First, we want to again stress that we included all publicly available studies released between May 2023 and December 2023 in the CELLxGENE census that used 10x Genomics sequencing and contained the same cell types represented in the training data, to ensure consistency with the training set. The resulting benchmark comprised 21 independent studies encompassing approximately 2.6 million human cells. The test set also includes cells collected from new tissues and disease contexts which are not present in the training data, providing a more challenging evaluation of model generalization. We make this explicit in our manuscript.

Lines 56-62: ***To better evaluate generalization to newly released studies, we consider an out-of-distribution (OOD) setup in which models are tested on datasets not seen during training (Figure 1b). We trained three methods with increasingly complex architectures (a linear classifier, a multilayer perceptron (MLP), and TabNet (Arik and Pfister 2021)) on an atlas of 15.2 million human cells annotated***

with 164 unique cell types, curated in the scTab study (Fischer et al. 2024) from the May 2023 release of the CELLxGENE census (Figure 1c). We then evaluate each method on 2.6 million human cells from 21 studies newly added during the 2023-12-15 release, spanning 470 donors, 16 tissues, and 80 of the original 164 cell types represented in the training set.

Lines 135-148: The dataset used in this study originates from the same filtered subset of the CELLxGENE census (version 2023-05-15) (Abdulla et al. 2024) that was curated for the scTab study (Fischer et al. 2024). This subset was constructed by applying strict inclusion criteria to the full census: only primary human cells profiled with 10x Genomics technologies were retained and the feature space was limited to 19,331 human protein-coding genes. Cell types were required to appear in at least 5,000 cells drawn from a minimum of 30 donors. All gene expression profiles were size-factor normalized to 10,000 counts per cell and log-transformed with a pseudocount of 1 (that is $f(x) = \log(x + 1)$). The resulting dataset included 22,189,056 cells annotated with 164 distinct cell types, spanning 5,052 donors and 56 tissues. For the in-distribution (ID) task, we adopted the same donor-partitioned data split used in Fischer et al. (2024) --- that is, 15,240,192 cells for training, 3,500,032 for validation, and 3,448,832 for testing. The out-of-distribution (OOD) test dataset consisted of all newly added human cells in a subsequent release of the CELLxGENE census (version 2023-12-15). These cells were also profiled using 10x Genomics platforms and annotated with one of the 164 labels observed during training. This resulted in approximately 2.6 million cells drawn from 21 studies, covering 80 of the 164 training cell types.

In response to the reviewer's concern, we acknowledge that additional complexity may arise when a new, previously unseen study requiring cell type annotation employs a different sequencing technology. For example, if the training data exclusively used 10x Genomics sequencing, but the test set included studies utilizing Smart-seq2, this could introduce additional variation. We did not investigate how such technological differences might impact the hierarchical or standard cross-entropy in our current work. Our primary aim was to isolate the effect of using an ontology-aware loss function, without confounding results due to batch effects which may influence model architectures in non-uniform ways. Nonetheless, we agree that this is an important and interesting direction, and we plan to explore it in future research.

Reviewer #2 (Remarks on code availability):

I did not have any critiques about the code previously and so did not do any new review of it. However, I have looked through the updated documentation in the GitHub provided in response to reviewer 1's critiques and agree it is nicely done and a valuable addition.

We thank the reviewer for the kind words and for the thorough review of our work.